

# Phenotype characteristics of gastric epithelial mucus in patients with different gastric diseases: from superficial gastritis to gastric cancer

Nannan Dong[1,2,3], Rui Guo[1,2,3], Yuehua Gong[1,2,3] and Yuan Yuan[1,2,3]

[1] The First Hospital of China Medical University, Key Laboratory of GI Cancer Etiology and Prevention in Liaoning Province, Shenyang, LiaoNing, China

[2] The First Hospital of China Medical University, Tumor Etiology and Screening Department of Cancer Institute and General Surgery, Shenyang, LiaoNing, China

[3] The First Hospital of China Medical University, Key Laboratory of Cancer Etiology and Prevention in Liaoning Education Department, Shenyang, LiaoNing, China

## ABSTRACT

**Background**. Gastric gland mucin is important for maintaining the basic function of the gastric mucosa, protecting it from foreign substances and reducing the occurrence of gastric diseases. Exploring the phenotype of gastric gland mucus changes during the progression of gastric disease is of great clinical significance.

**Methods**. A total of 483 patients with different gastric diseases were collected in this study, including 82 superficial gastritis (SG), 81 atrophic gastritis (AG), 168 dysplasia (GD), and 152 gastric cancer (GC). Mucin staining was performed using HID-ABpH2.5-PAS method and was further grouped according to the mucin coloration.

**Results**. The phenotypic characteristics of mucin during disease progression were divided into neutral, acidic, and mucus-free types. Furthermore, acidic mucus can be divided into type I, type II, and type III. The SG group was dominated by neutral mucus (100%), and the AG was dominated by acid mucus (81.48%), which gradually increased with the severity of atrophy ($P < 0.05$). The GD and GC groups were dominated by mucus-free (43.45%, 78.29%), and as the degree of GD worsened, neutral and acidic mucus gradually decreased and mucus-free increased ($P < 0.001$). From the SG, AG, GD, and GC progression, neutral and acidic mucus gradually decreased, and mucus-free gradually increased. Acidic mucin revealed that type III (red-brown black) mucin was predominant in AG, GD, and GC, and increased with the degree of AG, GD, as well as the biological behavior of GC. In the lesion adjacent to high-grade GD or GC, type III acid mucin is predominant.

**Conclusion**. There were three mucin phenotypes in the process of gastric diseases. With the disease progression, the trend of phenotypic change was that neutral and acidic mucus gradually decreased and mucus-free increased. The appearance of type III mucin suggested a relatively serious phase of gastric diseases and may be a more suitable candidate for follow-up monitoring of patients with GC risk.

Corresponding authors
Yuehua Gong, yhgong@cmu.edu.cn
Yuan Yuan, yuanyuan@cmu.edu.cn

## INTRODUCTION

Mucins, a family of large and heavily glycosylated proteins, represent the main component of this hydrophilic gel-like mixture, and act as key molecules in the maintenance of gastrointestinal homeostasis (*Duarte et al., 2016*; *Jin et al., 2017*) and protect the gastric mucosa from the damage from external environment (*Ota & Katsuyama, 1992*; *Amieva & El-Omar, 2008*). These mucins are divided into two subtypes: surface mucin and gland mucin (*De Bolos, Garrido & Real, 1995*; *Teixeira et al., 2002*). The former is secreted from the surface cells of the gastric mucosa, while the latter is located in the lower layer of the gastric mucosa from the gland mucous cells. The stomach wall covered with surface mucus will not be directly exposed to the gastric juice environment with strong acid and proteases. Under normal circumstances, surface mucin can reduce gastric mucosal irritation caused by food friction, resist pepsin damage to gastric mucosa and reduce $H^+$ penetration to maintain the mucus-bicarbonate barrier function (*Jin et al., 2017*). Surface mucus is also important in drugs absorption process, namely for those that are gastric pH-sensitive (*Virili et al., 2019*). Gastric gland mucins are derived from pyloric glands, mucus neck cells, and fundus gland cells (*Robbe et al., 2004*). These mucus are important for maintaining the basic function of the gastric mucosa, protecting against foreign substances, and reducing the occurrence of gastric diseases (*Nakayama, 2014*).

The occurrence of GC is due to the chronic inflammation of the gastric mucosa under the long-term effect of pathogenic factors, which can further cause AG, IM (intestinal metaplasia), GD and GC (*Oue et al., 2019*) following the Correa's Model. Exploring the changes of mucus phenotype during the above pathological process is of great significance to reveal the occurrence and development of gastric diseases. It has been reported that expression and glycosylation of mucins are associated with gastric carcinogenesis including invasion, proliferation and regulation of tumor cells (*Hollingsworth & Swanson, 2004*; *Andrianifahanana, Moniaux & Batra, 2006*; *Betge et al., 2016*; *Pinho & Reis, 2015*). Several studies have suggested that the type of mucin expressed in early and advanced gastric carcinomas is of clinical significance, alluding to biologic differences in precursor lesions and/or pathways of malignant transformation (*Ha Kim et al., 2006*; *Kim et al., 2013*; *Koseki et al., 2000*; *Yamazaki et al., 2006*). Accumulating evidence has indicated that different mucin phenotypes of GC have distinct clinical characteristics and exhibit specific genetic and epigenetic changes (*Oue et al., 2015*). Thus, mucin phenotype classification is useful to understand GC pathogenesis. However, it is not exactly clear how the phenotype of gastric gland mucus changes during the progression of gastric disease.

In this study, we used AB-PAS mucus staining method to explore the changes of mucus phenotype during the occurrence and development of gastric diseases, which would provide clues and help to reveal the occurrence and development of gastric diseases (*Oue et al., 2019*).

## MATERIALS AND METHODS

### The subjects

A total of 483 patients with different gastric diseases were included in this study, among them 283 patients underwent gastroscopic biopsy and 118 patients underwent endoscopic submucosal dissection (ESD) surgery at the Endoscopy Center of the First Affiliated Hospital of China Medical University from July 2014 to March 2018. In addition, 82 cases from the on-site GC screening project in Zhuanghe county, Liaoning province from January 2014 to December 2014. They were diagnosed as 82 SG, 81 AG, 168 GD, 152 GC, including 299 males and 184 females, with an average age of 60.25 years old. The *H. pylori* infection was detected using ASSURE *Hp* rapid test (MP Diagnostics$^{TM}$). The histopathological diagnostic criteria of the subjects referred to the updated Sydney classification system. This study was approved by the Ethics Committee of the First Affiliated Hospital of China Medical University, and the subjects signed informed consents ([2013]135 and 2016[161]).

### HID-ABpH2.5-PAS mucin histochemical staining

The specific steps of HID-ABpH2.5 PAS mucin histochemical staining followed the instructions of the kit (kit was purchased from Beijing Regen Biotechnology Co., Ltd.). In short, after dewaxing to water, the slides were placed in the solution (A1:A2 = 50:3), away from light for 18–24 h at 24 °C–28 °C. After that, stained with Alcian blue for 20 min, and next incubated with periodic acid for 30 s, and then schiff reagent was added for 2 min, and at last the slides were acidified, dehydrated and mounted. Neutral mucus material is red, acidic mucus material is blue, mixed mucus material is blue-purple or purple-blue, and the nucleus is brown.

### The phenotypes of different acid mucus

In acid mucus, sialic acid mucus is blue, and sulfuric acid mucus is brownish black. Further, according to the different coloration of acid mucin, the gastric mucosa is divided into type I (blue or brown-black), type II (red-blue) and type III (red-brown black). If there is more than one mucus phenotype in the lesion, the most important mucus phenotype is classified.

### Statistical analysis

SPSS 16.0 software was used for statistical analysis. $P < 0.05$ was statistically significant. The descriptive statistics of continuous variables are mean, standard deviation, minimum and maximum, while categorical variables are count and percentage. Chi-square test or Fisher's exact test were used to determine the statistical association between categorical variables, and t test was used to determine the statistical association between continuous variables.

## RESULTS

### The baseline characteristics of the subjects

A total of 82 SG, 81 AG, 168 GD, and 152 GC were included in this study, the average age and gender composition of each group were shown in Table 1, of which the SG group was younger compared with other groups, with statistical significance ($P = 0.000$). In terms

**Table 1** Baseline characteristics of 483 subjects.

| Characteristcs | SG | AG | GD | GC | *P* value |
|---|---|---|---|---|---|
| Total cases | 82 | 81 | 168 | 152 | |
| Age (years) | 52.09 | 61.46 | 61.73 | 62.39 | 0.000 |
| Sex (%) | | | | | |
| Male | 42 (51.22) | 46 (56.79) | 112 (66.67) | 99 (65.13) | 0.067 |
| Female | 40 (48.78) | 35 (43.21) | 56 (33.33) | 53 (34.87) | |

of male and female composition, the difference between the groups was not statistically significant ($P = 0.067$).

## Mucous phenotype characteristics of different gastric diseases

We compared the mucin staining of mucosal tissues in different gastric diseases, as shown in Table 2 and Fig. 1. Of the total 483 patients, 135 (27.89%), 154 (31.82%), and 195 (40.29%) were neutral, acidic mucus, and mucus-free respectively. Among them, 100% of the SG group was neutral mucus. The patients with AG were mainly acid mucus, accounting for 81.48%, followed by neutral mucus accounting for 14.81%, and the difference between the groups was statistically significant ($P = 0.000$). The patients with GD were mainly without mucus (43.45%), followed by acidic mucus (35.12%) and neutral mucus (21.43%), the difference between the groups was statistically significant ($P = 0.000$); patients with GC were similar to patients with GD, and mainly stained without mucus (78.29%), the rest were neutral mucus (2.63%) and acid mucus (19.08%). The difference between the groups was statistically significant ($P = 0.000$).

In the comparison between different disease groups, the expression of neutral mucus gradually decreased from SG, AG to GD and GC, with a positivity of 100%, 14.81%, 21.43% and 2.63%, respectively, and the difference between the groups was statistically significant ($P = 0.000$). The expression of acidic mucus gradually decreased (81.48%, 35.12% and 19.08%) respectively, the difference between the groups was statistically significant ($P = 0.000$). Those mucus-free expression gradually increased (3.71%, 43.45% and 78.29%) respectively, the difference between the groups was statistically significant ($P = 0.000$). It can be seen that SG was mainly composed of neutral mucus, AG was mainly expressed by acidic mucus, and GD and GC tissues were mainly expressed by mucus-free. With the progress of gastric disease, neutral and acidic mucus expression gradually decreased, and those without mucus expression gradually increased.

## Subgroup analysis of mucous phenotype characteristics of different gastric diseases

Further, we analyzed mucin staining in different subgroups of AG, GD, and GC. The results were shown in Table 3. In AG, there was no statistically significant difference in the percentage of neutral mucus and acid mucus in the antrum, corpus and angle of stomach ($P = 0.874$). In mild atrophy, neutral and acid mucus accounted for 33.33 and 53.33%, respectively. The acid mucus of moderate and severe AG were 85.11% and 94.74%, respectively, higher than that of mild AG, and the difference between the groups

**Table 2   Mucous phenotype characteristics of different gastric diseases.**

| Diseases | Cases | Neutral mucus (%) | Acidic mucus (%) | Non-mucus (%) | *P* value |
|---|---|---|---|---|---|
| Total | 483 | 135 (27.89) | 154 (31.82) | 195 (40.29) | |
| SG | 82 | 82 (100) | 0 | 0 | 0.000 |
| AG | 81 | 12 (14.81) | 66 (81.48) | 3 (3.71) | 0.000 |
| GD | 168 | 36 (21.43) | 59 (35.12) | 73 (43.45) | 0.000 |
| GC | 152 | 4 (2.63) | 29 (19.08) | 119 (78.29) | 0.000 |
| *P* value | | 0.000 | 0.000 | 0.000 | |

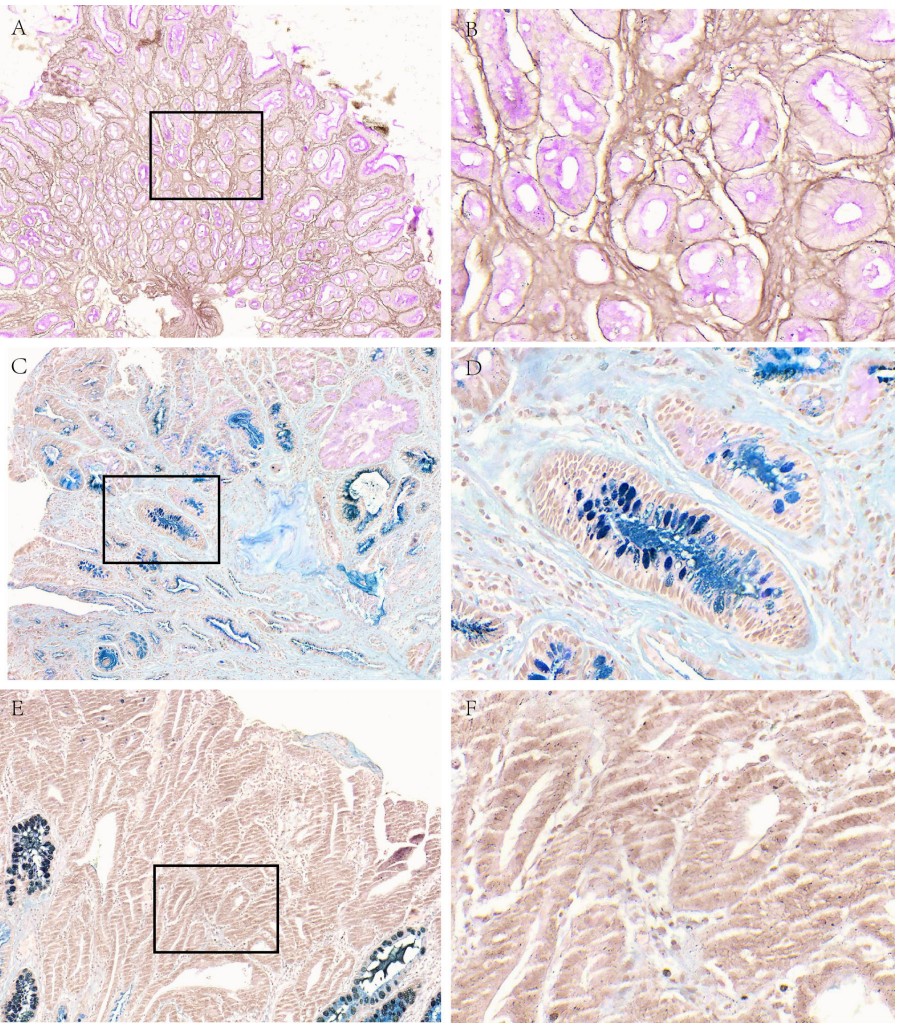

**Figure 1   Mucous phenotype characteristics of different gastric diseases.** (A, B) Representative picture of neutral mucus (red) in the SG group. (C, D) Representative picture of acid mucus (blue) in the AG group. (E, F) Representative picture of mucus-free in GC group. A, C and E were made at ×100 magnification; B, D and F were magnified images (×400) of the boxed sections depicted at left.

**Table 3  Subgroup analysis of mucous phenotype characteristics of different gastric diseases.**

| Diseases | Cases | Neutral mucus (%) | Acidic mucus (%) | Non-mucus (%) | *P* value |
|---|---|---|---|---|---|
| **AG** | 81 | 12 (14.81) | 66 (81.48) | 3 (3.71) | |
| **Position** | | | | | |
| Corpus | 15 | 2 (13.33) | 13 (86.67) | 0 (0) | |
| Angle | 13 | 2 (15.38) | 10 (76.92) | 1 (7.69) | 0.874 |
| Antrum | 53 | 8 (15.09) | 43 (81.13) | 2 (3.78) | |
| **Degree** | | | | | |
| Mild | 15 | 5 (33.33) | 8 (53.33) | 2 (13.34) | |
| Moderate | 47 | 6 (12.77) | 40 (85.11) | 1 (2.12) | 0.020 |
| Severe | 19 | 1 (5.26) | 18 (94.74) | | |
| **GD** | 168 | 36 (21.43) | 59 (35.12) | 73 (43.45) | |
| Low-grade | 96 | 33 (34.38) | 48 (50.00) | 15 (15.62) | 0.000 |
| High-grade | 72 | 3 (4.17) | 11 (15.28) | 58 (80.56) | |
| **GC** | 152 | 4 (2.63) | 29 (19.08) | 119 (78.29) | |
| Intestinal type | 78 | 4 (5.13) | 12 (15.38) | 62 (79.49) | |
| Diffuse type | 46 | 0 (0) | 11 (23.91) | 35 (76.09) | 0.500 |
| Mixed type | 3 | 0 (0) | 1 (33.33) | 2 (66.67) | |
| Unclassifiable | 25 | 0 (0) | 5 (20.00) | 20 (80.00) | |

was statistically significant ($P = 0.02$). In the low-grade of GD, neutral, acidic mucus, mucus-free were 34.38%, 50.00%, 15.62%, and in the high-grade were 4.17%, 15.28%, 80.56%, the difference between the groups was statistically significant ($P = 0.000$). In GC, mucus-free was the highest in intestinal-type, followed by diffuse-type, unclassifiable type and mixed-type, but the difference was not statistically significant ($P = 0.5$). It can be seen that in AG, acid mucus gradually increases with the severity of gastric disease, in GD, as the degree increasing, neutral and acid mucus gradually decreased, and no mucus increased. Among various histological types of GC, those with no mucus were the highest.

## The analysis of mucous phenotype characteristics of different *H. pylori* status

Among the AG, GD, and GC samples included in this study, there were 151 patients with *H. pylori* infection information. We compared the relationship of *H. pylori* infection status and different mucus phenotypes in the overall disease group and among the three disease groups, respectively. As shown in Table 4, the results indicated that in the overall disease group, there was no difference in mucus phenotype between the *H. pylori* positive and negative groups ($P = 0.724$). However in AG subgroup, the *H. pylori* positive group was mainly acidic mucus type (72.97%), and the *H. pylori* negative group was mainly neutral mucus (100%). There was no difference in mucus phenotype between *H. pylori* positive and negative groups in GD, GC subgroups ($P = 0.879, 0.819$).

## The distribution of acidic mucus phenotype in different diseases

We further conducted a comparative analysis of acidic mucus phenotypes among different gastric diseases, including AG, GD and GC, as shown in Table 5 and Fig. 2. The results

**Table 4  Subgroup analysis of mucous phenotype characteristics of different *H. pylori* status.**

| Diseases | Hp | Cases | Neutral mucus (%) | Acidic mucus (%) | Mucus-free (%) | *P* value |
|---|---|---|---|---|---|---|
| Total | Positive | 97 | 15 (15.46) | 41 (42.27) | 41 (42.27) | 0.724 |
| | Negative | 53 | 7 (13.21) | 20 (37.73) | 26 (49.06) | |
| AG | Positive | 37 | 8 (21.62) | 27 (72.97) | 2 (5.41) | 0.000 |
| | Negative | 8 | 8 (100) | 0 (0) | 0 (0) | |
| GD | Positive | 33 | 6 (18.18) | 10 (30.30) | 17 (51.52) | 0.879 |
| | Negative | 26 | 6 (23.08) | 8 (30.77) | 12 (46.15) | |
| GC | Positive | 27 | 1 (3.71) | 4 (14.81) | 22 (81.48) | 0.819 |
| | Negative | 19 | 1 (5.26) | 4 (21.05) | 14 (73.69) | |

**Table 5  The distribution of acidic mucus phenotype in different diseases.**

| Diseases | Cases | Type I (%) | Type II (%) | Type III (%) | *P* value |
|---|---|---|---|---|---|
| **CAG** | 66 | 11 (16.67) | 10 (15.15) | 45 (68.18) | 0.000 |
| **Position** | | | | | |
| Corpus | 13 | 0 (0) | 2 (15.38) | 11 (84.62) | |
| Angle | 10 | 1 (10) | 2 (20) | 7 (70) | 0.348 |
| Antrum | 43 | 10 (23.26) | 6 (13.95) | 27 (62.79) | |
| **Degree** | | | | | |
| Mild | 8 | 3 (37.5) | 1 (12.5) | 4 (50) | |
| Moderate | 40 | 5 (12.5) | 7 (17.5) | 28 (70) | 0.505 |
| Severe | 18 | 3 (16.67) | 2 (11.11) | 13 (72.22) | |
| **GD** | 59 | 3 (5.08) | 12 (20.34) | 44 (74.58) | 0.000 |
| Low-grade | 48 | 3 (6.25) | 12 (25) | 33 (68.75) | 0.014 |
| High-grade | 11 | 0 (0) | 0 (0) | 11(100) | |
| **GC** | 29 | 2 (6.90) | 6 (20.69) | 21 (72.41) | 0.000 |
| Intestinal type | 12 | 0 (0) | 3 (25) | 9 (75) | |
| Diffuse type | 11 | 0 (0) | 2 (18.18) | 9 (81.82) | 0.091 |
| Mixed type | 1 | 0 (100) | 0 (0) | 1 (100) | |
| Unclassifiable | 5 | 2 (40) | 1 (20) | 2 (40) | |

showed that the acidic mucus phenotype can be further subdivided into three subtypes: type I, sialic acid or mucin sulfate; type II, neutral and acidic sialic acid mucin, occasionally, mucin sulfate, or both; type III, neutral and sulfuric acid mucus mainly, occasionally, sialic acid mucin, or both.

Among 66 cases of AG, there was 11 cases of type I (16.67%), 10 cases of type II (15.15%) and 45 cases of type III (68.18%); in 59 cases of GD, there was 3 cases of type I (5.08), 12 cases of type II (20.34%) and 44 cases of type III (74.58%); in terms of 29 cases of GC, there were 2 cases of type I (6.9%), 6 cases of type II (20.69%) and 21 cases of type III (72.41%).

Further stratified according to location and degree in AG, there was no statistically significant difference in mucin subtype ($P = 0.348$, $P = 0.505$). In the 59 cases of GD, including 48 cases of low-grade and 11 cases of high-grade, type III were the mainly

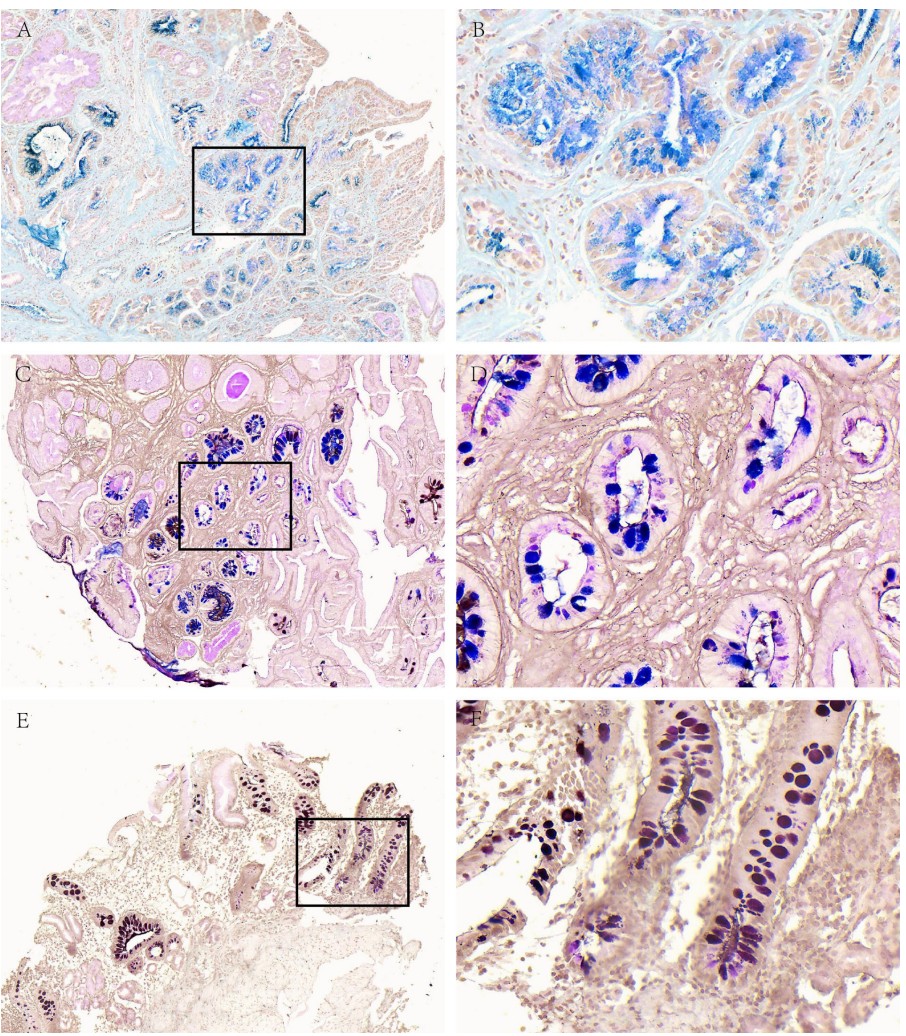

**Figure 2  Acidic mucus phenotype in AG.** (A, B) Representative picture of type I (blue). (C, D) Representative picture of type II (red-blue). (E, F) Representative picture of type III (red-brown black). A, C and E were made at ×100 magnification; B, D and F were magnified images (×400) of the boxed sections depicted in A, C and E.

subtype, especially with 100% in the high-grade ($P = 0.000$). There were 12 cases of intestinal-type GC and 11 cases of diffuse-type GC with mucin subtype, all of which were mainly type III, and the difference between the groups was not statistically significant ($P = 0.091$).

## The comparation of acid mucus phenotypes in the lesion adjacent to high-grade GD or GC

There were 112 cases of IM nearby the above-mentioned high-grade GD and GC. In order to further analyze the correlation between different mucus phenotypes and high-grade GD or GC. As shown in Table 6 and Fig. 3, in high-grade GD, intestinal and diffuse GC, type

**Table 6  The comparison of acid mucus phenotypes in the lesion adjacent to high-grade GD or GC.**

| GD/GC | Cases | Type I (%) | Type II (%) | Type III (%) | *P* value |
|---|---|---|---|---|---|
| High-grade | 42 | 4 (9.53) | 3 (7.14) | 35 (83.33) | 0.000 |
| Intestinal type | 49 | 4 (8.16) | 1 (2.04) | 44 (89.8) | |
| Diffuse type | 13 | 1 (7.69) | 0 (0) | 12 (92.31) | 0.980 |
| Mixed type | 2 | 0 (0) | 0 | 2 (100) | |
| Unclassifiable | 6 | 1 (16.67) | 0 | 5 (83.33) | |

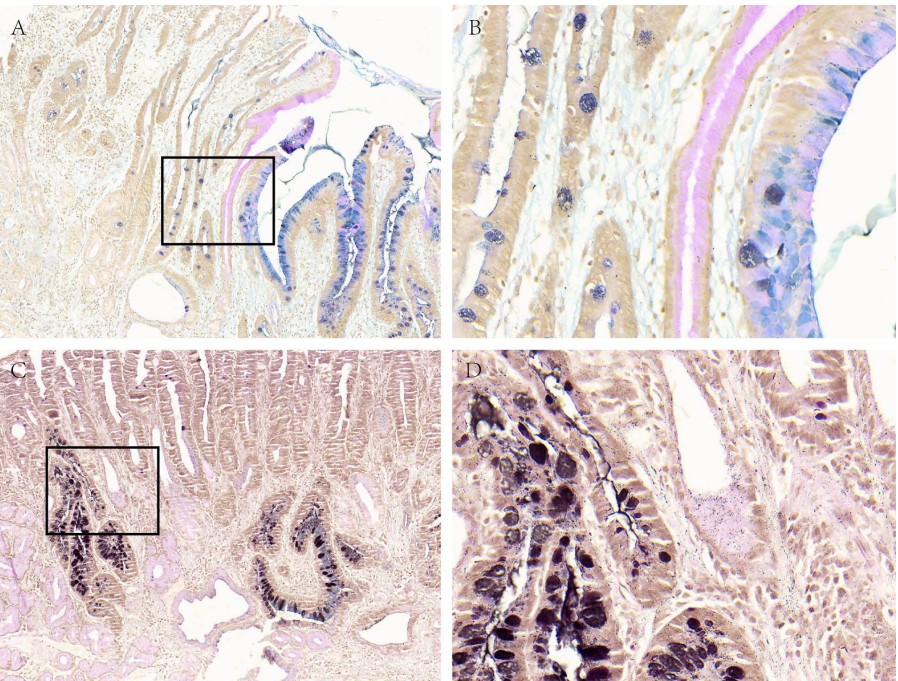

**Figure 3  Acid mucus phenotypes in the lesion adjacent to GC.** (A, B) Acid mucus of type II can be seen in the lesion adjacent to GC. (C, D) Acid mucus of type III can be seen in the lesion adjacent to GC. A and C were made at ×100 magnification; B and D were magnified images (×400) of the boxed sections depicted in A and C.

III acid mucin is predominant, without statistically significant difference between groups ($P = 0.714$).

## DISCUSSION

Current research findings suggest that mucin subtypes and clinical relevance are reported differently in the literature, but most of the data always support that different mucin subtypes are associated with the risk of intestinal gastric adenocarcinoma (*Ha Kim et al., 2006*; *Kim et al., 2013*; *Koseki et al., 2000*; *Yamazaki et al., 2006*). That is to say, the progression of gastric precancerous lesions to GC is a slow and gradual development process, at least targeted monitoring can be performed by mucin typing, and there is an opportunity to detect and remove tumorous lesions early. Although there are currently no

clinically deterministic predictive biomarkers in clinical use, the use of mucin staining for subtype analysis can further stratify people who are already at risk (*Hondo et al., 2017*). In this study, we explored the changes of different mucus phenotypes during the progression of gastric disease.

Usually, the gastric epithelium, pyloric gland, and duodenal gland mainly secrete neutral mucus (*Linden et al., 2008*). The goblet cells and intestinal glands of the small and large intestine mucosa mainly secrete acidic mucus (*Reis et al., 1999*). Under normal physiological conditions, the staining of gastric mucin showed red neutral mucin, but in some pathological conditions, such as IM, carcinogenesis, etc., the characteristics of gastric mucin staining can change with the specific state. Kazuhiro Yamanoi et al. reported that the reduction of gastric mucus is related to the high mitotic activity of tumor cells, which represents an increase in malignant potential (*Yamanoi & Nakayama, 2018*). The results of our study showed that in the SG, neutral mucus was dominant, but in AG with IM, acid mucus was mainly expressed, which produced by intestinal metaplasia glands. In the GD and GC, it was mainly expressed as mucus-free phenotype. During the progression of gastric diseases from SG, AG, GD, and GC, the expression of neutral and acidic mucus gradually decreased, mucus-free expression gradually increased. From this we can speculate that the disappearance of epithelial mucus phenotype may be one of the high-risk factors for the progression of gastric disease.

*H. pylori* persistent infection of the normal gastric mucosa triggers a chronic inflammatory process designated by chronic gastritis. The presence of virulent *H. pylori* strains together with host immune vulnerability can lead to severe mucosal atrophy with focal loss of gland architecture and disease progression (*Correa & Houghton, 2007*). In fact, the development of IM originates multiple foci where superficial foveolar cells with neutral mucin expression are gradually replaced by acidic producing cells with an intestinal phenotype (*Correa, 1992*). In our study, we found in AG subgroup, the *H. pylori* positive group was mainly acidic mucus type and the *H. pylori* negative group was mainly neutral mucus. There was no difference in mucus phenotype between *H. pylori* positive and negative groups in GD, GC subgroups. It was the truth that with the progress of IM, the colonization of *H. pylori* gradually decreased, especially in the stage of GC, there was basically no colonization (*Chen et al., 2019*), so AG was dominant with acid mucus in *H. pylori* positive.

The main feature of intestinal epithelium is acid mucin staining while once IM occurs in the gastric mucosa, acidic mucin can be shown in gastric epithelium. Acid mucin can be divided into sialic acid or sulfated mucin, which can be used to evaluate two types using combined HID-AB staining, and the latter is brown stained with ferric diamine (HID). *Filipe et al. (1994)* used this method to distinguish mucin phenotypes: type I expressed only salivary mucin, type II expressed a hybrid form of a mixture of gastric mucosa and intestinal mucin, and type III expressed neutral and sulfated mucin. It is still not clear whether there is a chronological relationship between the expression of these three types of mucins (*Reis et al., 1999*; *Silva et al., 2002*; *Gutierrez-Gonzalez & Wright, 2008*). During the initial development of gastric diseases with IM, the neutral mucin in normal mucosa gradually decreased, while salivary mucin appeared and became the main type of mucin.

In the serious stages of IM, sulfate mucin appeared and may become the main ingredient. In this study, we detected acid mucin in AG, GD, and GC, and found that type III mucin was the main type in the three types of lesions, and increased with the change of the IM degree, the GD grade and the biological behavior of GC. In lesion next to high-grade GD, intestinal-type and diffuse GC, the type III acidic mucin is predominant. The emergence of type III mucin is a progressive stage of IM, which can become the best candidate for follow-up monitoring of patients with GC risk. In addition, based on a meta-analysis of 7 studies, including patients with GIM in 2014 and no accompanying dysplasia (929 patients with type II or III mucin, and 1112 patients with type I mucin), these patients ranged from 3-12.8 years of GC developed during the follow-up period. Compared with individuals of type I, the risk of GC was 3.33 times higher (95% confidence interval, 1.96–5.64) (*Altayar et al., 2020*; *Gawron et al., 2020*). Consistent with the AGA (American Gastroenterological Association) guideline (*Banks et al., 2019*), the recent ESGE (European Society of Gastrointestinal Endoscopy) guideline also mentioned that mucin staining has prognostic value, which is contrary to the 2012 ESGE guideline that opposes GIM subtype guidance for prognosis (*Dinis-Ribeiro et al., 2012*; *Pimentel-Nunes et al., 2019*). Considering the potential prognostic value and the opportunity to generate stronger clinical and epidemiological evidence, as well as the pathologist's minimum cost and effort, recommendations for routine typing of mucin are advisable.

From a histochemical point of view, the pH value of normal gastric mucin is neutral, and they are dyed red with periodate-Schiff (PAS). Chronic AG is histologically characterized by chronic inflammation of the gastric mucosa, with decreasing number of gastric gland cells and increasing intestinal metaplastic glands, therefore acid mucin replaces the original gastric mucin and gets blue stained with pH 2.5 Alcian blue. Therefore, AB-PAS combined staining can distinguish normal epithelium with IM. Recently, the expression profile of different mucins has been analyzed with antibodies. MUC5AC, MUC6, MUC2, and CD10 have been found to be specifically expressed in the gastric foveolar epithelium, pyloric gland cells, goblet cells, and brush border, respectively (*Kim et al., 2013*). However, this antibody staining method cannot further distinguish GI mixed type mucin into salivary mucin and sulfated mucin, that is, it cannot distinguish between type II and type III. Moreover, compared with antibody staining, the HID-AB-PAS method is more time-saving, efficient, and cheaper and cost-effective.

However, the current study was limited that although we have seen the dynamic changes of gastric mucus phenotype during the progression of gastric disease, whether the disappearance of mucus phenotype is "cause" or "effect", the conclusions are not consistent, and further study is needed in the future such as functional assays with relevant study models (cell lines, mice…). Although we found that *H. pylori* positive and negative groups have different dominant mucus phenotype, however, in AG, GD, GC subgroups, the difference was not obvious. And due to the limitation of the number of cases, we did not further analyze the acidic mucus subtypes between *H. pylori* status, which has yet to be further studied.

In summary, this study explored the phenotype of gastric gland mucus changes during the progression of gastric disease and found that the disappearance of mucus phenotype

is one of the high-risk factors for the progression of gastric diseases. The appearance of type III mucin is a relatively serious phase of gastric diseases and may be a more suitable candidate for follow-up monitoring of patients with GC risk. In China, there is currently still no effective method for clinical management and prognosis follow-up of patients with IM and dysplasia. With the continuous improvement of our understanding of mucin typing, the expansion of research samples and the increasing awareness of clinicians, we will get more evidence to support, continue to improve our clinical management of GIM and dysplasia patients, and ultimately improve the early detection rate of GC and reduce its mortality.

### Funding
This work was supported by the National Natural Science Foundation of Liaoning Province (2019-MS-388). The funders had no role in study design, data collection and analysis, decision to publish, or preparation of the manuscript.

### Grant Disclosures
The following grant information was disclosed by the authors:
National Natural Science Foundation of Liaoning Province: 2019-MS-388.

### Competing Interests
The authors declare there are no competing interests.

### Author Contributions
- Nannan Dong performed the experiments, prepared figures and/or tables, authored or reviewed drafts of the paper, and approved the final draft.
- Rui Guo analyzed the data, prepared figures and/or tables, and approved the final draft.
- Yuehua Gong conceived and designed the experiments, analyzed the data, prepared figures and/or tables, authored or reviewed drafts of the paper, and approved the final draft.
- Yuan Yuan conceived and designed the experiments, analyzed the data, authored or reviewed drafts of the paper, and approved the final draft.

### Human Ethics
The following information was supplied relating to ethical approvals (i.e., approving body and any reference numbers):
Our study was approved by the Ethics Committee of the First Affiliated Hospital of China Medical University (approval number: [2013]135) and (ethics numbers 2016[161]).

### Data Availability
Raw data, including mucous phenotype characteristics of different gastric diseases and acidic mucus phenotype in CAG and in the lesion next to GC, are available as Supplemental File.

## Supplemental Information

Supplemental information for this article can be found online at http://dx.doi.org/10.7717/peerj.10822#supplemental-information.

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
