# Peer review of "Phenotype characteristics of gastric epithelial mucus in patients with different gastric diseases: from superficial gastritis to gastric cancer"

_PeerJ, doi:10.7717/peerj.10822_

## Round 0.1 · original submission · Major Revisions

There are important points to address as indicated by the Reviewers. The acceptance of the manuscript depends on how the authors will answer to each of them. Please pay attention also to English editing!

Reviewer 1 ·

Basic reporting

English language needs a professional editing
Background and context in the field are ok
Structure is ok
Relevant results are self- contained

Experimental design

Everything is ok

Validity of the findings

Everything is Ok

Additional comments

This is an interesting manuscript that clearly shows that variation in mucus composition characterizes different gastric disorders and different grades of gastric mucosa damage.
Beside the need of a clerical language revision, I have few comments:

- all your data are addressed to elucidate the progression from atrophy to gastric cancer, that is perfect to me. However, mucus is important also in drugs absorption process, namely for those that are gastric pH-sensitive (Lahner E, et al. Thyro-entero-gastric autoimmunity: Pathophysiology and implications for patient management. Best Pract Res Clin Endocrinol Metab. 2020;34(1):101373, Sigurdsson HH, et al. Mucus as a barrier to lipophilic drugs. Int J Pharm. 2013;453(1):56-64). Among these drugs oral thyroxine is on the spot due to the large number of treated patients worldwide and the narrow therapeutic index (see for rev Virili C, et al. Gastrointestinal Malabsorption of Thyroxine. Endocr Rev. 2019;40(1):118-136. I believe that this topic should be quoted in the paper.

- Discussion section contains several repetitions of data and should be shortened as well as integrated by the topic suggested above. The role of gastric mucus in different pathophysiologic areas should be highlighted and the related references added.

- Tab 4 is redundant and the data are already in the text.

Minor points
- Intestinal metaplasia and not its acronym (IM) should be cited the first time it appears (line71)
- line 61: alkaline mucin is cited only once. Is this equivalent to the neutral mucin or is a mistake? Please clarify.
- line 156 is GAG acronym correct?

Reviewer 2 ·

Basic reporting

The introduction section is somewhat vague and reads more like a textbook on the subject. Only 6 references are provided. Authors should provide examples in the literature to support their hypothesis on the putative role of different mucus phenotypes on cancer progression. There are multiple studies evaluating both mucin expression and their glyco-profile and correlation with clinical-pathological features in the gastric context that could be cited.

The histochemical images in the figures should be improved for clarity. Besides the overall view of the lesion, insets with higher-magnification should be provided in these figures.

English revision is advised throughout the manuscript.

Lines 72-74, Introduction – “Exploring the changes of mucus phenotype during the above pathological changes is of great significance to reveal the occurrence and development of gastric diseases”. Could the authors provide evidence in the literature that supports this sentence?

Lines 227-229, Discussion – “From this we can speculate that the disappearance of epithelial mucus phenotype is one of the high-risk factors for the progression of gastric disease.” This is highly speculative without providing some functional prove that there is cause-effect.

Minor points
Line 71, Introduction – define IM abbreviation in full as it is the first time it is mentioned.

Lines 124-126, Results – Authors should provide the exact P value for the different statistical tests, not just the indication that it is over or below 0.05. This applies to all other sections in the manuscript.

Experimental design

An appropriate number of clinical cases is analysed. It is not clear why the authors did not analyse staining in IM cases with the description of the corresponding mucous phenotype characteristics, although these are sparsely mentioned in the main text. Also, Helicobacter pylori infection status is not taken into account in the analysis.

Validity of the findings

The association between loss of differentiation features (in this case presence of mucus-producing cells) and cancer progression is a well-known feature. Overall, as there is progression from pre-neoplasic lesions to increasingly aggressive tumours, differentiation features are lost. In the particular case of gastric cancer, this is well documented, and the expression profile of different mucins has been analysed with antibodies, which is more specific than histochemical methods that only provide information about the nature of the mucous (cf. Reis et al., 1999; Senapati et al., 2008; Namikawa and Hanazaki, 2010; Boltin and Niv, 2013). Although the findings are valid, they are not entirely new. The added benefit to the body of knowledge already published is not clear.

Additional comments

The way the results are reported could be improved for clarity. It is not clear in the end what is the main conclusion of the manuscript. With this type of study (associative in nature) one cannot extrapolate conclusions as to whether the mucous phenotype itself is involved in disease progression or merely a bystander effect, due to shifts in cellular phenotype. This type of conclusion warrants functional assays with relevant study models (cell lines, mice…).

Reviewer 3 ·

Basic reporting

No comment

Experimental design

No comment

Validity of the findings

No comment

Additional comments

The manuscript by Dong et al describe the changes in gastric mucus characteristics in different lesions. This is a subject addressed in different studies, that here is described is addressed in a very clear manner, using a robust sample of patients. the results are relevant and interesting yet there some issues that need to be addressed:

- designations could be more intuitive: for instance superficial gastritis should be SG and not GS; atrophic gastritis why is CAG?
- in the introduction sentence in lines 76-77 is speculative. Should be toned doen.
- mucins produced in atrophic gastritis are produced by intestinal metaplasia glands. This should be better described.
-The discussion needs to be streamlined and better written.
- some of the images have artifacts and should be improved.

---

## Round 0.2 · accepted · Accept

The paper has been revised properly.

Reviewer 1 ·

Basic reporting

Manuscript has been improved and my questions were met. No further comments.

Experimental design

Same as above

Validity of the findings

Same as above

Additional comments

No further comments